# Lineage does not regulate the sensory synaptic input of projection neurons in the mouse olfactory bulb

**Luis Sánchez-Guardado, Carlos Lois\***

Division of Biology and Biological Engineering, California Institute of Technology, Pasadena, United States

**Abstract** Lineage regulates the synaptic connections between neurons in some regions of the invertebrate nervous system. In mammals, recent experiments suggest that cell lineage determines the connectivity of pyramidal neurons in the neocortex, but the functional relevance of this phenomenon and whether it occurs in other neuronal types remains controversial. We investigated whether lineage plays a role in the connectivity of mitral and tufted cells, the projection neurons in the mouse olfactory bulb. We used transgenic mice to sparsely label neuronal progenitors and observed that clonally related neurons receive synaptic input from olfactory sensory neurons expressing different olfactory receptors. These results indicate that lineage does not determine the connectivity between olfactory sensory neurons and olfactory bulb projection neurons.
DOI: https://doi.org/10.7554/eLife.46675.001

## Introduction

The relationship between cell lineage and neuronal connectivity in the brain is not well understood. Lineage regulates the synaptic connections between neurons in some regions of the invertebrate nervous system. For example, in the *Drosophila* olfactory system, projection neurons are specified by cell lineage to receive synaptic input from the axons of specific types of olfactory sensory neurons (OSNs) (*Jefferis et al., 2001*; *Li et al., 2018*). In mammals, it has been reported that clonally related pyramidal neurons are preferentially connected to each other in the neocortex (*Yu et al., 2009*; *Yu et al., 2012*; *He et al., 2015*). Furthermore, it has been proposed that sister neurons in the visual cortex have a strong correlation to the stimuli to which they respond (*Li et al., 2012*), while other works suggest that this correlation is much weaker (*Ohtsuki et al., 2012*). To further investigate the role played by lineage in the assembly of brain circuits we focused on the mammalian olfactory bulb, a brain region with an anatomical organization particularly advantageous to study this question.

The mammalian olfactory system can be divided into three regions: olfactory epithelium, olfactory bulb (OB) and olfactory cortex. The olfactory epithelium harbors the OSNs. Each OSN expresses just one of more than one thousand odorant receptors (*Buck and Axel, 1991*; *Chess et al., 1994*). OSN axons expressing the same odorant receptor converge into one or two discrete neuropil structures in each OB called glomeruli, forming a stereotypic map on the OB surface (*Ressler et al., 1994*; *Vassar et al., 1994*; *Mombaerts et al., 1996*; *Wang et al., 1998*). The projection neurons in the OB are called mitral and tufted cells (M/T cells). In mammals, the majority (>90%) of M/T cells have a single apical dendrite that branches into a single glomerulus (*Mori, 1987*; *Shepherd, 1990*; *Malun and Brunjes, 1996*) where they receive sensory input from OSNs expressing a particular odor receptor (*Figure 1A*) (*Ressler et al., 1994*; *Vassar et al., 1994*; *Stewart et al., 1979*; *Mori, 1987*; *Malun and Brunjes, 1996*; *Matsutani and Yamamoto, 2000*). Thus, the anatomical organization of the glomerulus in the OB is an ideal system to investigate the possible relationship between lineage and connectivity because the apical dendrite of the M/T cells provides a direct readout of their synaptic

**\*For correspondence:**
clois@caltech.edu

**Competing interests:** The authors declare that no competing interests exist.

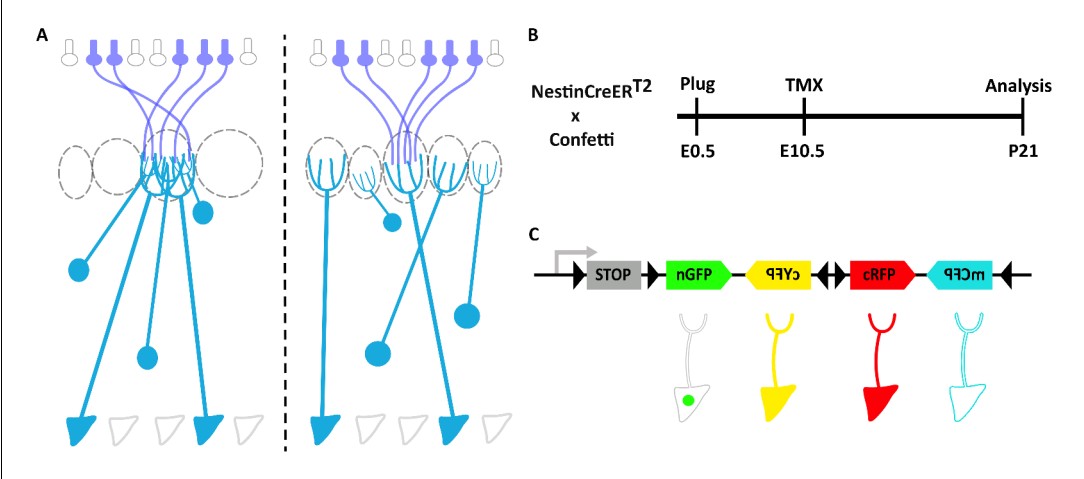

**Figure 1.** Clonal analysis of projection neurons using *Nestin-CreER^T2^::Confetti* mice to sparsely label neuronal progenitors. (**A**) Schematic representation of the olfactory bulb (OB). Axons from olfactory sensory neurons (OSNs) expressing the same receptor project to a single glomerulus, forming synaptic contacts with the apical dendrites of mitral and tufted cells. Two possible scenarios of the relationship between lineage and connectivity are presented. (left) The apical dendrites of clonally related M/T cells innervate the same glomerulus, indicating that lineage regulates their connectivity. (right) The apical dendrites of sister M/T cells innervate different glomeruli, indicating that connectivity of M/T cells is independent of their lineage (**B**) Experimental design to label neuronal progenitors with tamoxifen (TMX) at embryonic day 10 (E10.5), and their posterior analyses at postnatal day 21 (P21). (**C**) The *Confetti* cassette encodes four different fluorescent proteins (nuclear GFP (nGFP), membrane CFP (mCFP), and cytoplasmic YFP (cYFP) and RFP (cRFP)). Upon Cre recombination, the STOP sequence is excised and randomly expressed one out four possible fluorescent proteins.

DOI: https://doi.org/10.7554/eLife.46675.002

The following figure supplements are available for figure 1:

**Figure supplement 1.** M/T and pyramidal neurons labeled with different fluorescent proteins.
DOI: https://doi.org/10.7554/eLife.46675.003
**Figure supplement 2.** Labeling of progenitor cells at E10.5 in a *Nestin-Cre::Ai9* mouse.
DOI: https://doi.org/10.7554/eLife.46675.004
**Figure supplement 3.** M/T cells labeled at P7 in a *Nestin-Cre::Ai9* mouse.
DOI: https://doi.org/10.7554/eLife.46675.005

input. To address this question, we sparsely labeled M/T cells progenitors and investigated the sensory input that their progeny receives from OSNs. Our results show that sister M/T cells receive synaptic input from different glomeruli, indicating that lineage does not determine the sensory input of the OB projections neurons, and suggest that the connectivity between OB projection neurons and olfactory sensory neurons depends on other mechanisms, including random targeting of dendrites towards glomeruli and activity-dependent mechanisms.

## Results and discussion

### Labeling of progenitors of OB projection neurons

The projection neurons in the OB are called mitral and tufted cells (M/T cells). M/T cells originate from progenitors located in the OB primordium, which is derived from the rostral part of the dorsal telencephalon (*Hinds, 1968a*; *Hinds, 1968b*). To investigate the lineage of M/T cells, we crossed two transgenic mouse: *Nestin-CreER^T2^* (*Kuo et al., 2006*), which can be used to activate Cre in neuronal progenitors in a sparse manner, and *Confetti* (*Snippert et al., 2010*), which can label individual cells with one out four possible fluorescent proteins upon Cre-mediated recombination (*Figure 1B,C* and *Figure 1—figure supplement 1*) (*Kuo et al., 2006*; *Snippert et al., 2010*).

To investigate whether Nestin promoter drives Cre recombinase activity into M/T cell progenitors, we crossed the driver *Nestin-Cre* mouse (*Tronche et al., 1999*) with the reporter *Ai9* mouse (*Madisen et al., 2010*) and confirmed the labeling both of OB progenitors in the embryo, and M/T cells in the adult (*Figure 1—figure supplement 2* and *Figure 1—figure supplement 3*). To be able

to perform clonal analysis, we optimized the conditions to label just a handful of progenitors, ideally a single progenitor per OB. First, we confirmed that our transgenic mice *Nestin-CreER^{T2}::Confetti* did not label any neurons in the brain without tamoxifen (TMX) administration (n = 3; data not shown). Second, we found that with an injection of 1 mg of TMX per 40 g of body weight into a 10-day pregnant female (E10.5) we observed a handful of labeled pyramidal neuron clones in the neocortex, and around 20 M/T cells labeled in the OB when the brains were examined at postnatal day 21 (P21) (*Figure 1B* and *Figure 1—figure supplement 1*). Third, we confirmed that this TMX concentration labeled a few progenitors per brain when animals were analyzed 2 days after TMX administration (E12.5) (*Figure 2*). With these conditions, we observed between none to a single progenitor labeled per fluorescent protein in the OB primordium (n = 6 embryos), the presumptive location of the M/T progenitors. Although we observed a very low number of progenitors labeled, we cannot unambiguously conclude whether a group of cells labeled at P21 with the same fluorescent protein in the OB originated from a single progenitor or two independent progenitors. However, because of the low number of clones labeled with these conditions we will work under the assumption that any group of M/T cells labeled with the same fluorescent protein in the OB are part of a single clone.

To study the lineage of the M/T cells we induced Cre activity at E10.5, the peak time for mitral cell generation (*Hinds, 1968a*; *Hinds, 1968b*; *Blanchart et al., 2006*; *Kim et al., 2011*; *Imamura et al., 2011*). Brains were analyzed at P21, once M/T cells have completed the refinement of their dendrites and they have a mature morphology with a single apical dendrite projecting into a single glomerulus (*Figure 1A*) (*Malun and Brunjes, 1996*; *Lin et al., 2000*; *Matsutani and*

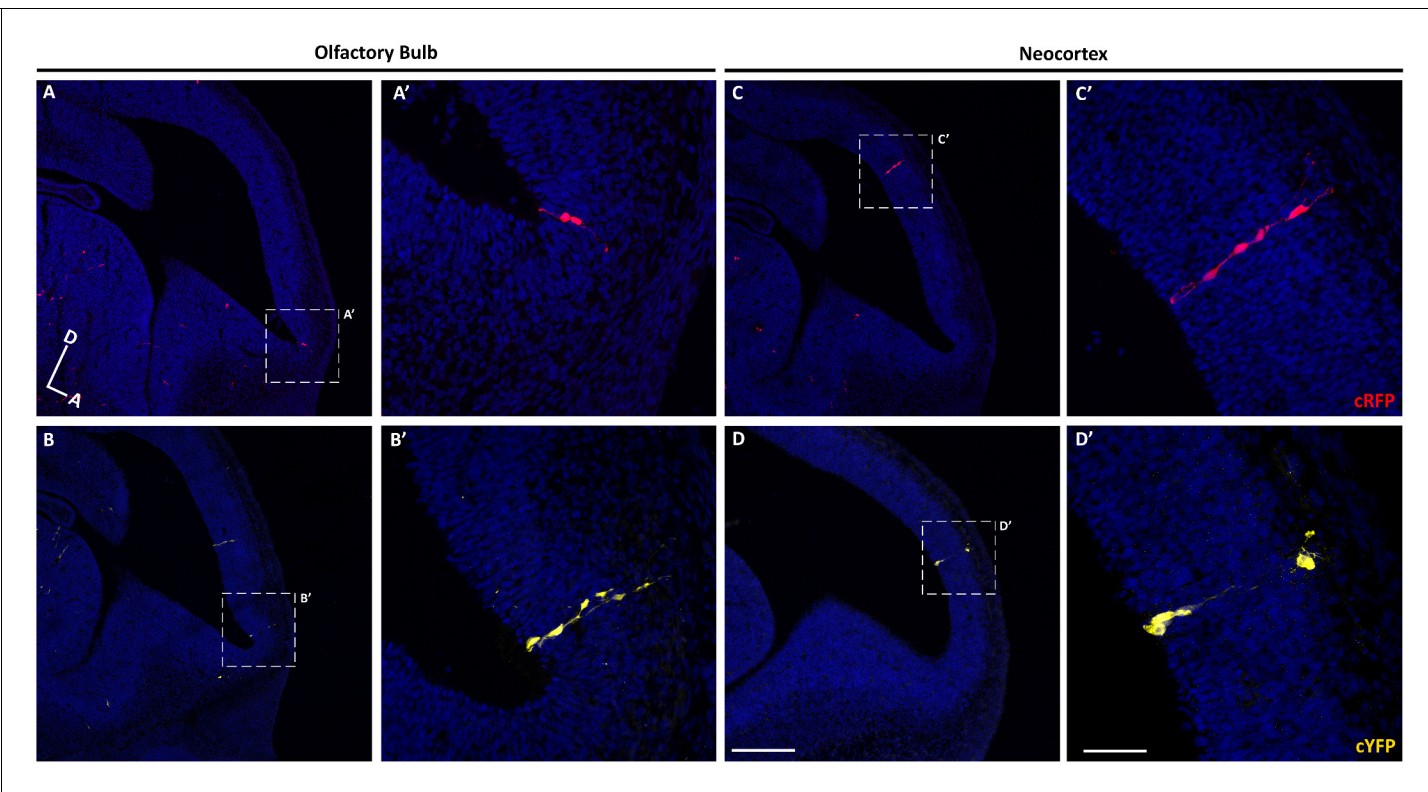

**Figure 2.** Sparse labeling of progenitor cells in the embryonic mouse brain. (**A–D**) Sagittal sections through the brain of an E12.5 mouse treated with TMX at E10.5. (**A–B**) Confocal images of individual clones labeled in the OB expressing cRFP (**A–A'**) and cYFP (**B–B'**). (**A'–B'**) High-magnification images of the clones shown in A and B. (**C–D**) Single clones labeled in the neocortex expressing cRFP (**C–C'**) and cYFP (**D–D'**). (**C'–D'**) High-magnification images of the clones shown in **C–D**). Cell nuclei are labeled with DAPI (blue). Scale bar in D is 200 µm and applies to A-D, scale bar in D' is 50 µm applies to A'-D'. Orientation of brains: D, dorsal; A, anterior.

DOI: https://doi.org/10.7554/eLife.46675.006

The following figure supplement is available for figure 2:

**Figure supplement 1.** Progenitor cells labeled in neocortex with three different fluorescent proteins.

DOI: https://doi.org/10.7554/eLife.46675.007

*Yamamoto, 2000*; *Blanchart et al., 2006*). *Confetti* mice can produce four different fluorescent proteins with distinct subcellular locations (cytosolic (cRFP and cYFP), membrane (mCFP), and nuclear (nGFP)) (*Figure 1C*, *Figure 1—figure supplement 1* and *Figure 2—figure supplement 1*) (*Snippert et al., 2010*). Consistent with previous works, we observed that many clones in the OB were labeled by RFP (n = 9), whereas YFP (n = 4) and CFP (n = 1) clones appeared less frequently (*Reeves et al., 2018*). We did not analyze any of the nGFP+ cells for two reasons. First, the most reliable way to unambiguously identify M/T cells is by their distinctive morphology. Nevertheless, if a cell is only labeled in the nucleus (as in nGFP+ cells), we cannot tell apart M/T cells from other OB cell types (e.g. short axon cells, granule cells, juxtaperiglomerular). Second, to identify the connectivity between M/T cells and glomeruli, it is necessary to follow the projection of their apical dendrites (*Figure 1—figure supplement 1*), and we cannot observe any dendrites in the nGFP+ cells.

In total, we analyzed 28 OBs. 15 of them did not have any labeled M/T cells. 11 OBs had both M and T cells labeled (n = 14 clones). Of these 11 OBs, eight had putative clones of a single color, and the remaining three OBs had two clones labeled with different fluorescent proteins. Two OBs had clones that contained only M cells (n = 2 clones). We do not know the reason why these two OBs had only M cells labeled, and several reasons may account for this observation, including progenitors committed to produced only M cells. We did not find any OB with only T cells labeled when TMX was administered at E10.5.

## Size of clones and distribution of neurons in the OB and neocortex

We measured the putative clone size in the OB and compared them with neocortex clones. We found 310 labeled M/T cells in 14 putative clones in the OB, such that the average OB clone contained 22.14 ± 6.61 M/T cells (average ± standard deviation). We found 556 labeled cells in six neocortex clones, such that the average cortical clone contained 92.67 ± 23.18 pyramidal neurons (average ± standard deviation), consistent with previous results (*Franco et al., 2012*; *Gao et al., 2014*) (*Figure 3A*). These observations suggest that the clone size in the neocortex is approximately four times larger than in the OB, consistent with the previous results (*Cárdenas et al., 2018*).

We analyzed the distribution of the cell bodies of the labeled M/T cells in the 14 clones containing M and T labeled cells in the OB (n = 310 neurons) and labeled pyramidal neurons in the six neocortex clones (n = 556 neurons) by performing 3D reconstructions using the Neurolucida software (*Figure 3B–D*, *Figure 3—figure supplement 1* and *Figure 3—figure supplement 2*). The 3D reconstructions revealed that sister M/T cells were distributed in a broader area than the tight columns of sister pyramidal neurons in the neocortex (*Figure 3—figure supplement 1* and *Figure 3—figure supplement 2*). To analyze the distribution of cells from each clone, we calculated the nearest neighbor distance (NND) based on the distances of neurons in our 3D reconstructions (*Figure 3E* and *Figure 3—figure supplement 3*). We found that sister M/T cells were more separated from each other (287.47 μm ± 61.23; average ± standard deviation) than sister pyramidal neurons (59.56 μm ± 9.86) (*Figure 3E*). The dispersion of sister M/T cells that we observed is consistent with the tangential migration of immature M/T cells reported in the embryonic OB (*Blanchart et al., 2006*; *Imamura et al., 2011*).

To investigate whether the distribution of sister M/T cells observed was random, we compared the NNDs of the labeled M/T cells observed (n = 310) with a simulated random dataset. The same strategy was followed for neocortex clones. We found that the NNDs between clonally related neurons were shorter than the simulated random datasets both for the OB and neocortex (*Figure 3E*, $p<0.01$; two-way ANOVA). Similar results were reported for pyramidal neurons in the neocortex (*Gao et al., 2014*). This indicates that although sister M/T cells are not obviously clustered, their distribution in the OB is not random. Interestingly, previous works have observed that the tangential migration of immature M/T cells in the embryonic OB may be regulated by gradients of secreted (*Inokuchi et al., 2017*) or cell adhesion molecules (*Bastakis et al., 2015*), biasing their distribution to specific regions within the OB.

Previous experiments have demonstrated that migration of M/T cells is biased toward the dorsal or ventral regions of the OB at different developmental times (*Imamura et al., 2011*). In addition, it has been hypothesized that the dorsal and ventral domains of the OB may have a preference to process innate and learned odorants, respectively (*Kobayakawa et al., 2007*). To investigate whether the cell distribution in a clone was biased toward a specific OB domain, we divided the OB into two domains based on the expression of the OSN markers NQO1 and OCAM, that label the dorsal and

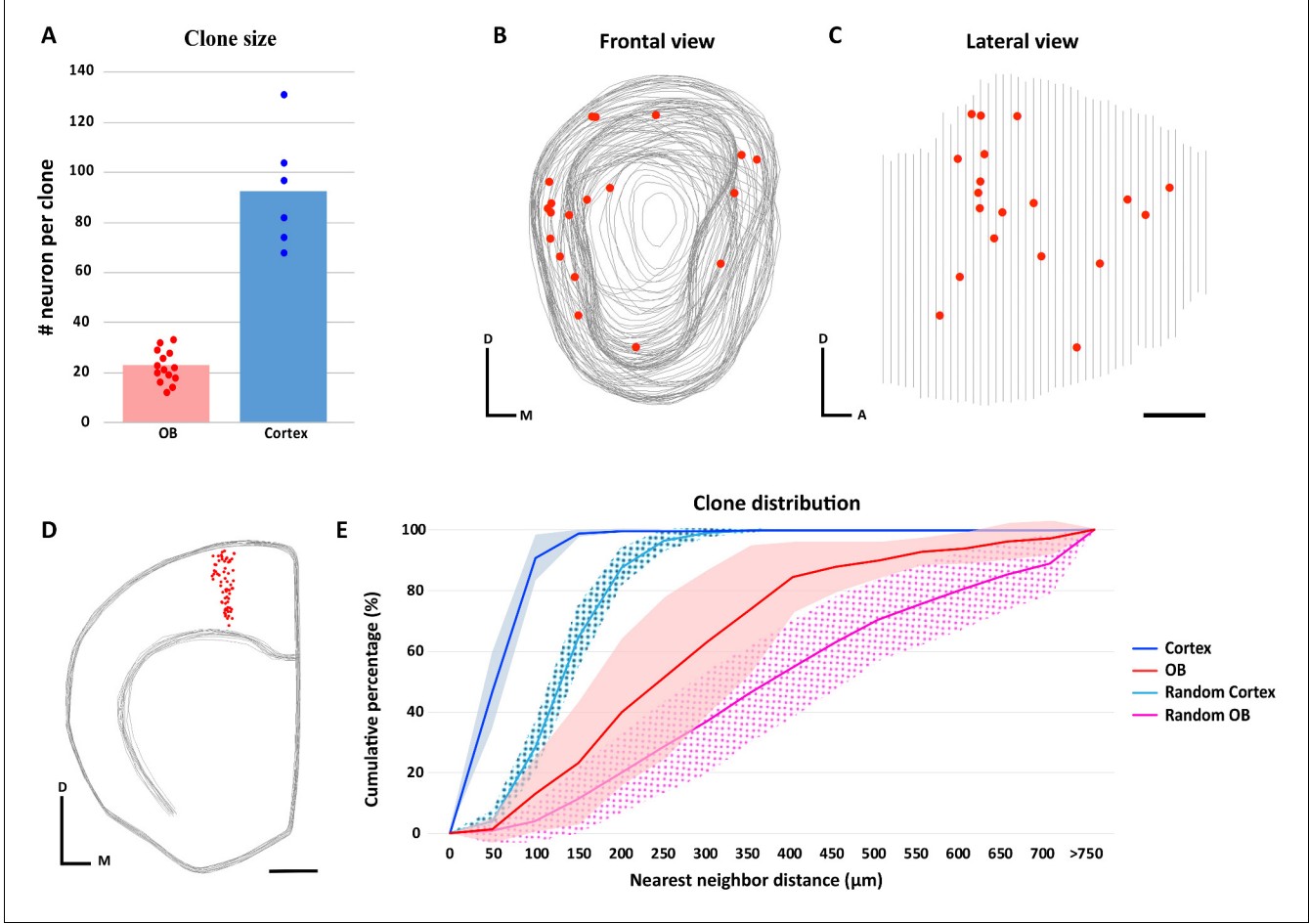

**Figure 3.** Clone size and distribution of cells labeled in the olfactory bulb and neocortex. (**A**) Clone size quantification in the OB and neocortex. Data are shown as average showing all data points. (**B–D**) 3D reconstruction of a *NestinCreER^T2^::Confetti* P21 mice OB (**B–C**) and neocortex (**D**) treated with TMX at E10.5. Gray lines indicate the contours of the brain and red dots represent the cell bodies of labeled neurons. (**B**) Frontal and (**C**) lateral views of the 3D reconstruction of one OB. (**D**) Frontal view of the neocortex 3D reconstruction. (**E**) Cumulative percentage of the NNDs of sister neurons labeled in the OB (red) and neocortex (dark blue). Data are shown as average ± standard deviation (OB, n = 310 neurons in 14 clones; neocortex, n = 556 neurons in six clones). Pink and light blue lines represent 100 datasets of random simulations of OB and neocortex NND, respectively (see also *Figure 3—source data 1*). No significant differences were observed when real OB clones were compared to different real OB clones, or when real neocortex clones were compared to different real neocortex clones (OB, p=0.96; neocortex, p=0.95; two-way ANOVA). However, significant differences were observed when real clones were compared with their respective simulated clones (for both OB and neocortex, p<0.01: two-way ANOVA). Scale bar in C is 0.5 mm and applies to B-C. Scale bar in D is 1 mm. Orientation of diagrams in B-D: D, dorsal; A, anterior; M, medial.
DOI: https://doi.org/10.7554/eLife.46675.008

The following source data and figure supplements are available for figure 3:

**Source data 1.** Quantification of the cell NNDs in real and randomized OB and neocortex clones at E10.5.
DOI: https://doi.org/10.7554/eLife.46675.013
**Source data 2.** Quantification of the cell NNDs in real OB and neocortex clones at E12.5.
DOI: https://doi.org/10.7554/eLife.46675.014
**Figure supplement 1.** 3D reconstruction of clones labeled in the olfactory bulb.
DOI: https://doi.org/10.7554/eLife.46675.009
**Figure supplement 2.** 3D reconstruction of clones labeled in the neocortex.
DOI: https://doi.org/10.7554/eLife.46675.010
**Figure supplement 3.** NND distribution of single clones based on their cell number.
DOI: https://doi.org/10.7554/eLife.46675.011
**Figure supplement 4.** Clone size and distribution of cells labeled in the olfactory bulb and neocortex when TMX was administered at E12.5.
DOI: https://doi.org/10.7554/eLife.46675.012

ventral regions of the OB, respectively (*Figure 3—figure supplement 1K*; *Gussing and Bohm, 2004*; *Yoshihara et al., 1997*). Then, we analyzed the distribution of clonally related M/T cells throughout these two domains (*Figure 3—figure supplement 1*). Of the 14 OB clones we analyzed, four clones had a bias toward the ventral OB domain, three clones for the dorsal domain, and the remaining seven clones had similar number of cells in the dorsal and ventral domains. Overall, when all the clones were analyzed together, there were no preferences in the distribution of M/T cells towards the dorsal or ventral domains (*Figure 3—figure supplement 1L*; p=0.67, t-test). Similarly, we did not detect any bias for the distribution of M/T cells OB in the lateral or medial domains (data not shown).

To analyze whether labeling of M/T progenitors at different developmental times could influence the distribution of M/T cells to a specific OB domain, we performed additional experiments to label M/T progenitors at a later time point by injecting TMX into 12 day pregnant females (E12.5), and brains were examined at P21, as in the E10.5 experiment. Previous works have demonstrated that in the neocortex the number of neurons per clone is reduced as progenitors are labeled at later embryonic stages (*Angevine and Sidman, 1961*; *Walsh and Cepko, 1988*; *Luskin et al., 1988*; *Price and Thurlow, 1988*; *Rakic, 1988*; *Gao et al., 2014*). Consistent with this observation, the clones that were labeled at E12.5 in the OBs contained fewer cells than at E10.5: 8.44 ± 6.37 M/T cells per clone (average ± standard deviation, n = 76 cells) when labeled at E12.5, compared with 22.14 ± 6.61 M/T cells per clone when labeled at E10.5 (*Figure 3—figure supplement 4B, C-I'*). Similarly, we observed a reduction in the number of cells per clone in the neocortex when labeling progenitors at later developmental stages (22.5 ± 6.47 pyramidal neurons (n = 135 cells) at E12.5 versus 92.67 ± 23.18 pyramidal neurons at E10.5)), consistent with previous results (*Franco et al., 2012*; *Gao et al., 2014*) (*Figure 3—figure supplement 4B,K–P*). In total we analyzed 18 OBs with progenitors labeled at E12.5. Eleven OBs did not have any M/T labeled cells. Seven OBs had nine clones with labeled M/T cells. Of these seven OBs, five s had a single putative clone, each clone labeled with a single fluorescent protein. Each of the other two OBs had two clones labeled with different fluorescent proteins. As in our E10.5 experiment, we observed that sometimes the cells in a clone were preferentially located in a specific domain (dorsal or ventral, or medial or lateral), although overall we did not find any significant differences in their distribution (*Figure 3—figure supplement 4J*).

Our experiments were designed to investigate the relationship between lineage and connectivity in the main olfactory bulb (MOB). Although not the primary goal of our work, these experiments gave us the opportunity to investigate whether M/T cells in the accessory olfactory bulb (AOB) were clonally related to the M/T cells in the MOB. When TMX was administered at E12.5, we did not find any M/T cells labeled in the AOB, consistent with the observation that AOB M/T cells are born earlier than MOB M/T cells (*Hinds, 1968a*). When we injected TMX at E10.5 we observed a small number of labeled M/T cells in the AOB. We inspected 28 OBs labeled at E10.5, and found that 10 OBs contained 18 M/T cells labeled in the AOB, with only 1–3 labeled M/T cells per AOB. Four OBs had 1–2 labeled M/T cells in the AOB and none in the MOB. Four OBs had M/T cells labeled with the same fluorescent proteins in both MOB and AOB, with only 1–3 cells in each AOB. The remaining two OBs had one cell in each AOB labeled with a fluorescent protein different from the M/T cells labeled in the MOB (see table in *Supplementary file 1*). Although these small numbers do not allow for a definitive conclusion, our results suggest that there are separate progenitors for the M/T cells in the MOB and AOB. This hypothesis is consistent with recent works indicating that some M/T cells in the AOB are born from progenitors located in the diencephalic-telencephalic boundary, which then migrate rostrally to the posterior AOB (*Huilgol et al., 2013*; *Ruiz-Reig et al., 2017*). Further experiments will be required to clarify these questions.

## Synaptic input of sister M/T cells

It has been proposed that the anatomical organization of the OB may be analogous to the neocortex columnar organization. In the neocortex it is thought that the pyramidal neurons forming part of a column perform a similar task (*Rakic, 1988*; *Mountcastle, 1997*). Similarly, M/T cells receiving synaptic input from the same glomerulus respond to the same odorant (*Kauer and Cinelli, 1993*; *Mori et al., 1999*; *Bozza et al., 2002*). Our results indicate that sister M/T cells are widely distributed throughout the OB (*Figure 3*, *Figure 3—figure supplement 1* and *Figure 3—figure supplement 4*). Based on this observation, it may seem unlikely that sister M/T cells would have apical

dendrites projecting into the same glomerulus. However, this could still be possible because the soma of M/T cells innervating the same glomerulus may be separated from each other up to 450 µm (for M cells) and 350 µm (for T cells) (*Liu et al., 2016*). To investigate whether sister M/T cells receive synaptic input from the same glomerulus, we tracked their apical dendrites (*Figure 4* and *Figure 4— figure supplement 1*). Among all the labeled M/T cells that we detected (310 cells from 14 putative M/T clones (E10.5) and 74 cells from nine putative M/T clones (E12.5)), we never observed two neurons innervating the same glomerulus, even when their cell bodies were near each other (*Figure 4B– E*, *Figure 4—figure supplement 1E*). Nevertheless, it is still possible that, although we did not observe them, there may exist clones of M/T cells in the OB genetically pre-determined to project to the same glomerulus. This scenario could be expected for putative glomeruli responsive to relevant odors for survival, such as those responsive to predators or poisons, which require an innate

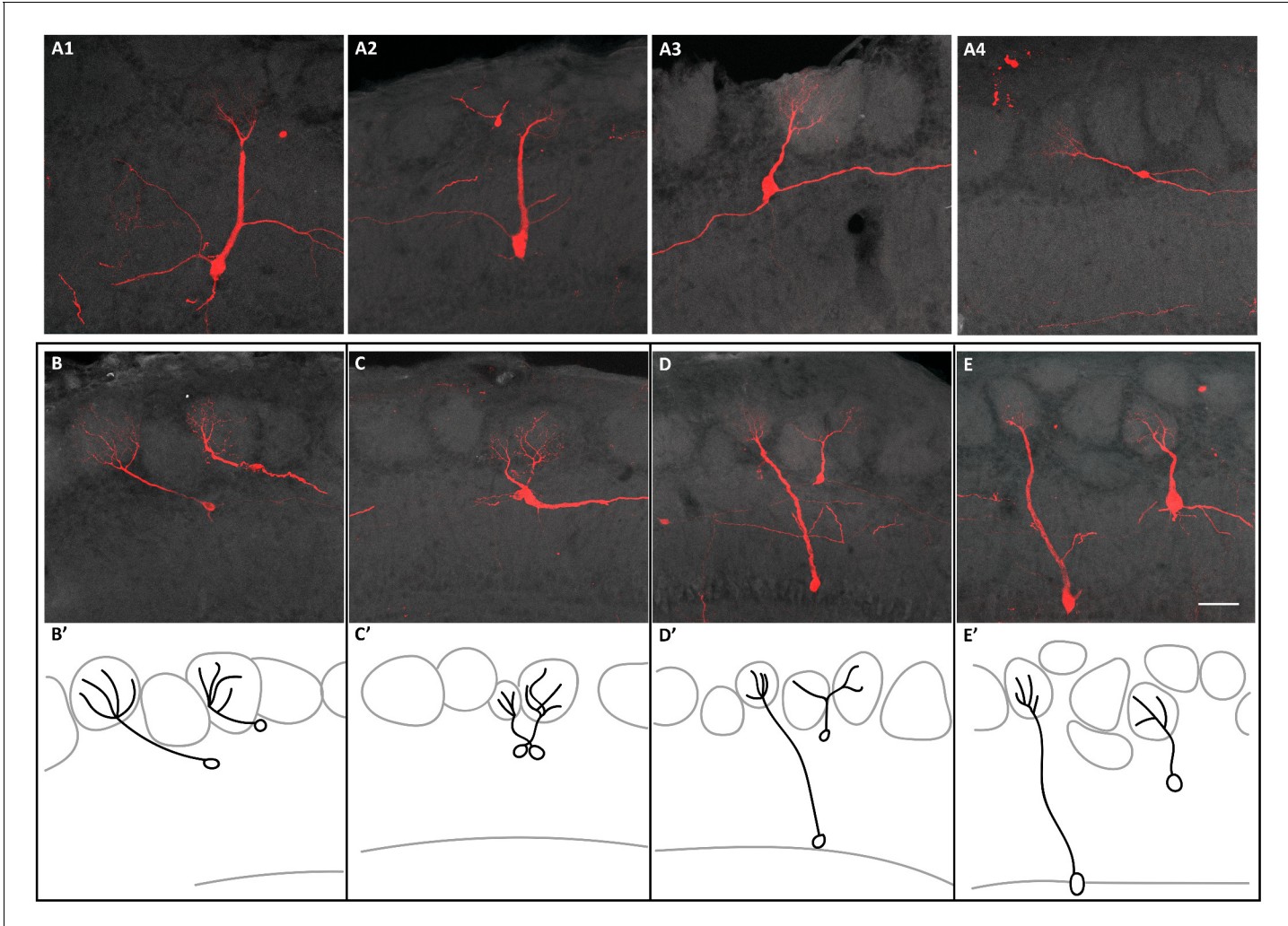

**Figure 4.** Connectivity of clonally related M/T cells when TMX was administered at E10.5. (**A**) Confocal images of four sister M/T cells belonging to a putative individual clone in the OB. (**B–E**) Confocal images of sister M/T cells from four clones, in four different OBs, with their somata close to each other and their apical dendrites innervating different glomeruli. (**B'–E'**) Schematic representation of the confocal images in B-E. Scale bar in E is 50 µm and applies to A-E.

DOI: https://doi.org/10.7554/eLife.46675.015

The following figure supplements are available for figure 4:

**Figure supplement 1.** Connectivity of clonally related M/T cells when TMX was administered at E12.5.
DOI: https://doi.org/10.7554/eLife.46675.016

**Figure supplement 2.** Connectivity of clonally related M/T cells in the AOB.
DOI: https://doi.org/10.7554/eLife.46675.017

and hardwired response of avoidance (*Kobayakawa et al., 2007*; *Sosulski et al., 2011*). Future experiments analyzing a much larger number of clones than those detected here may reveal the existence of these putative 'hardwired' M/T clones.

It is generally thought that the AOB has a preference for innate odorants, and thus, one may anticipate that lineage may regulate the connectivity of AOB projection neurons. However, there is a critical caveat that make it difficult to investigate the relationship between lineage and connectivity in the AOB. Although glomeruli are clearly distinct in the MOB, glomeruli in the AOB are less well defined and more difficult to identify. As indicated above, we observed only a small number of AOBs (four out of 10) that contained more than one (2 or 3) labeled M/T cells. Although the small number of labeled AOB M/T cells does not allow us to draw any firm conclusions, we did not find any M/T cells whose apical dendrites innervated the same glomerulus (*Figure 4—figure supplement 2*), similar to what we observed in the MOB.

In summary, our results indicate that lineage does not determinate the input connectivity of the apical dendrites of projection neurons in the mammalian OB. This is in contrast to what has been described for projection neurons in the *Drosophila* antennal lobe (*Jefferis et al., 2001*) and suggested for pyramidal neurons in the rodent visual cortex (*Li et al., 2012*). Our results indicate that the sensory input received by M/T cells is regulated by other factors independent of lineage, including random targeting of dendrites towards glomeruli and activity-dependent mechanisms, consistent with previous observations from multiple lines of evidence. First, during early postnatal stages M/T cells have several dendrites (between 3 to 5), and each of these dendrites project into different glomeruli that are close to each other, and immediately above their cell bodies (*Hinds, 1968a*; *Blanchart et al., 2006*). Starting approximately 1 week after birth, a process of refinement occurs such that around 90% of M/T cells retain just one apical dendrite and retract all others, and that remaining single apical dendrite branches into a single glomerulus (*Malun and Brunjes, 1996*; *Lin et al., 2000*; *Matsutani and Yamamoto, 2000*; *Blanchart et al., 2006*). It is important to note that even in full adult animals approximately 10% of mature M/T cells have two apical dendrites that project into two different glomeruli (*Lin et al., 2000*). Interestingly, the refinement by which M/T cells retain a single dendrite is a process partially dependent on neuronal activity. Olfactory deprivation by naris occlusion retards the refinement of M/T cell dendrites by approximately one week, although eventually the refinement process is accomplished to the same degree as in non-manipulated animals (*Matsutani and Yamamoto, 2000*). Interestingly, a recent work demonstrated that genetic blocking of action potentials in M/T cells prevented the dendrite refinement process such that even in adult animals the majority of M/T cells have several dendrites projecting into multiple glomeruli (*Fujimoto et al., 2019*). Finally, recent experiments indicate that activity-dependent mechanisms can direct the projection of M/T cell dendrites into specific glomeruli. For example, sensory odor experience in utero recruits the apical dendrites of M/T cells to the activated glomeruli (*Liu et al., 2016*). Similarly, genetic ablation of a large set of OSNs results in the absence of a large number of glomeruli in the dorsal OB, and in these animals, some M/T cells located in those regions lacking glomeruli extend their dendrites tangentially for a long distance until they reach a region with glomeruli, where they branch (*Nishizumi et al., 2019*).

In summary, multiple observations indicate that M/T cells are not committed to project into specific glomeruli. Instead, the available evidence, including the data presented here, suggests a model where progenitor cells give rise to a clone of sister M/T cells that migrate throughout the olfactory bulb such that sister cells disperse independently from each, and their cell bodies do not end up close to each other in specific regions of the bulb. After neuronal migration is completed, immature M/T cells initially grow multiple dendrites that receive synaptic input from multiple glomeruli without any apparent specificity. After a period of refinement regulated, in part, by neuronal activity, most (but not all) M/T cells retain a single dendrite that branches into a single glomerulus. However, the available evidence indicates that any of the multiple apical dendrites displayed by immature M/T cells can be retained, suggesting that M/T cells are not committed to receive synaptic input from any specific glomeruli. Finally, it is curious that the targeting of OSN axons and M/T dendrites toward the glomeruli appears to be regulated by very different mechanisms. Each OSN expresses a single olfactory receptor molecule that instructs its axons to project into a single glomerulus with high specificity (*Ressler et al., 1994*; *Vassar et al., 1994*; *Mombaerts et al., 1996*; *Wang et al., 1998*). In contrast, the existing evidence suggests that the apical dendrite of M/T cells can project to any glomeruli within a certain distance from the position of their cell bodies, without any apparent

specificity. What is the relationship between the OSN axons and the M/T dendrites for synapse formation in the glomeruli? Animals with mutations in the Tbr1 gene that result in the complete loss of M/T cells demonstrate that OSN axons can still reach the OB and converge into glomeruli-like structures in the same location as in wild-type animals (*Bulfone et al., 1998*). These experiments suggest that the targeting of OSN axons into the OB to form glomeruli does not require the presence of M/T cells. In contrast, the apical dendrites of M/T cells cannot form glomeruli in regions in mice in which a large set of olfactory receptors are genetically ablated, indicating that M/T cells require the presence of OSN axons to target their apical dendrites (*Kobayakawa et al., 2007*; *Nishizumi et al., 2019*).

Is there any biological advantage to the dispersion of projection neurons in the OB such that sister M/T cells receive synaptic input from different OSNs? Interestingly, it has been proposed that the M/T cells receiving input from the same glomerulus exhibit a wide diversity in their biophysical properties, and this diversity may be important for neural coding (*Padmanabhan and Urban, 2010*). In addition, neurons in the piriform cortex receive synaptic input from M/T cells innervating different glomeruli (*Miyamichi et al., 2011*), whereas M/T cells connected to the same glomerulus project their axons into many different areas of the olfactory cortex (*Sosulski et al., 2011*; *Ghosh et al., 2011*). However, the connectivity between M/T cells and the amygdala appears to be more stereotypical than between the M/T cells and other targets in the olfactory cortex (anterior olfactory nucleus, piriform cortex, tenia tecta, olfactory tubercle, cortical amygdala and entorhinal cortex) (*Haberly, 2001*; *Sosulski et al., 2011*). Based on these observations, one can speculate that the connectivity between the OB and its targets in the olfactory cortex may occur by two different mechanisms. Genetic factors, including lineage, may contribute to the connectivity between M/T cells and the amygdala, as this brain area is involved in innate behavior responses that may require hardwired connections (*Sosulski et al., 2011*). In contrast, the connectivity between M/T cells and areas of the olfactory cortex involved in the perception of odors that do not elicit innate behaviors are more plastic and may be regulated by mechanisms independent of lineage, such as random neurite targeting and activity-dependent wiring, among others (*Caron et al., 2013*; *Schaffer et al., 2018*). Our results indicating that lineage does not determine the sensory synaptic input of M/T cells raise further questions about the assembly of the olfactory circuits, including the mechanisms regulating the formation of synapses between OSNs and M/T cells, the role that experience may play sculpting the odor representations in the piriform cortex, and whether lineage regulates the connections with the amygdala to trigger innate behaviors.

## Materials and methods

### Animals

*Nestin-CreER^{T2}*, *Nestin-Cre*, *Confetti*, and *Ai9* mice were obtained from Jackson Laboratory. The *Nestin-Cre* and *Nestin-CreER^{T2}* mice can be used to induce the activity of Cre recombinase in neuronal progenitors directly or by the administration of tamoxifen (TMX) into animals, respectively (*Tronche et al., 1999*; *Kuo et al., 2006*). The *Ai9* mouse is a Cre-dependent reporter that expresses tdTomato fluorescent protein upon cre-mediated recombination (*Madisen et al., 2010*), while the *Confetti* mouse is a Cre-dependent reporter that produces four different fluorescent proteins (*Snippert et al., 2010*). We crossed the *Nestin-Cre* mouse with the *Ai9* mouse and the *Nestin-CreER^{T2}* mouse with the *Confetti* mouse. The resulting transgenic *Nestin-Cre::Ai9 and Nestin-CreER^{T2}::Confetti* mice were used for the experiments. For the timed pregnancy, the plug date was designated as E0.5 and the day of birth as P0. In all experiments, mice were handled according to the protocols approved by the Caltech Institutional Animal Care and Use Committee (IACUC). Mice colonies were maintained at the animal facility of the California Institute of Technology (Caltech).

### Tamoxifen induction

Tamoxifen (TMX, Sigma T-5648) was dissolved in 37°C pre-warmed corn oil (Sigma C8267) at a concentration of 10 mg/ml. *NestinCreER^{T2}::Confetti* embryos were induced at E10.5 (embryonic day 10.5) by a single intraperitoneal injection of 1 mg TMX into pregnant females (~40 grams). Animals were euthanized at embryonic day 12 (E12.5) or postnatal day 21 (P21).

## Tissue processing, immunohistochemistry, and imaging

Mouse embryos (E10.5 and E12.5) were fixed by immersion in 4% paraformaldehyde (PFA) in phosphate-buffered saline (PBS, pH 7.4) at 4°C overnight. Postnatal mice (P7 and P21) were fixed by intracardiac perfusion with 4% PFA in PBS. Brains were then extracted and incubated in 4% PFA at 4°C overnight. Next day, all samples were washed three times, 10 min each, with 0.1 M PBS, pH 7.4. Postnatal mice (P21) brains were embedded into 3% agarose and cut in a vibratome into 60 μm thick sections. Sections were collected sequentially. Embryonic and P7 brains were cut with a cryostat into 20 μm thick sections as previously described (*Sánchez-Guardado et al., 2009*).

We amplified the signal from fluorescent proteins by performing immunohistochemistry with antibodies against RFP and GFP. Although anti-GFP antibody recognizes nGFP, cYFP and mCFP proteins, we were able to distinguish between them based on the different subcellular location of the proteins (nuclear, cytoplasmic and membrane). In the figures, cells are shown with their original colors from the *Confetti* cassette, even though the signal from cYFP and mCFP proteins was amplified using the antibody against GFP (*Figure 1—figure supplement 1*, *Figure 2*, *Figure 2—figure supplement 1*). We did not include nGFP+ cells in our analyses because we cannot identify their morphology.

For immunocytochemistry, we incubated the sections for 30 min in blocking solution containing 1% bovine serum albumin in 0.1 M PBS-0.1% Triton X-100 (PBS-T). Sections were incubated overnight with the following primary antibodies diluted into blocking solution: 1:1000 chicken anti-GFP, Aves Labs Cat# GFP-1020 (RRID:AB_10000240), 1:1000 rabbit anti-RFP, Lifespan Cat# LS-C60076-100 (RRID:AB_1514409), 1:1000 rat anti-RFP, ChromoTek Cat# 5f8-100 (RRID:AB_2336064); 1:500 rat anti-Tbr2, Thermo Fisher Scientific Cat# 14-4875-82 (RRID:AB_11042577), 1:10,000 rabbit anti-Tbx21(kind gift from Y. Yoshihara), 1:250 rabbit anti-PAX6, Covance Cat# PRB-278P, (RRID:AB_291612), 1:20 mouse anti-RC2, DSHB Cat# RC2, (RRID:AB_531887). The next day, sections were washed three times, 10 min each, in PBS-T. Later, sections were incubated for 90 min at room temperature with secondary antibodies: Goat anti-chicken IgY Alexa-488 (RRID:AB_2534096), Donkey Anti-Rat IgG Alexa-488 (RRID:AB_2535794), Goat anti-Rabbit IgG Alexa-488 (RRID:AB_143165), Goat anti-Mouse IgG Alexa-488 (RRID:AB_2534069), Goat anti-Rat IgG Alexa-555 (RRID:AB_141733), Goat anti-Rabbit IgG Alexa-555 (RRID:AB_2535850), Goat anti-Rabbit IgG Alexa-647 (RRID:AB_2535812) diluted 1:1500 in blocking solution. Finally, the sections were counterstained with DAPI (D9542, Sigma), mounted sequentially on glass slides and mounted with Fluoromount (F4680, Fluoromount Aqueous Mounting Medium).

Z-stacks images were acquired using 10x, 20x or 40x objectives on a confocal microscope (Zeiss LSM 800). Z-stacks were merged and analyzed using ImageJ and edited with Photoshop (Adobe) software.

## 3D reconstruction and data analysis

Each section was analyzed and traced in sequential order from rostral to caudal using Neurolucida and StereoInvestigator software (MBF Bioscience Inc, Williston, VT). The boundaries of the OB and neocortex were traced and used to line up each section with the previous one to form 3D reconstructions. Each labeled cell in the OB or neocortex was tagged with a dot. Blue dots represent mCFP cells, red dots cRFP cells and green dots cYFP.

The distribution of the nearest neighbor distance (NND) was calculated using Matlab based on the cell coordinates of our 3D reconstruction created in Neurolucida software. NND was calculated by identifying the shortest straight path between labeled cells using the Euclidean distance. The NND was represented as cumulative percentage (average ± standard deviation) of the clones analyzed in the OB (n = 14) and neocortex (n = 6) (*Figure 3E*). In addition, we generated a dataset of random simulations based on the same number of M/T cells detected in our experiments (n = 310). The random dataset was generated based on the external plexiform layer (EPL) volume from one of the OBs analyzed. Using Matlab, we randomized eight times the number of cells of each OB clone in the EPL volume (n = 112 simulations). Using the same procedure, we randomized 17 times the number of pyramidal neurons of each neocortical clone (n = 102 simulations). The volume of one neocortex clone, representative of the average, was used as a volume boundary. Data are presented as average ± standard deviation, and statistical differences in the clone distribution were determined using two-way analysis of variance (ANOVA).

The division of the OB into dorsal and ventral domains was based on the expression of the NQO1 and OCAM markers (*Figure 3—figure supplement 1*) based on the previous published results (see Figure 7 in *Cho et al., 2007* and Figure 1 in *Imamura et al., 2011*). The results were analyzed using unpaired two-tailed t-test.

## Acknowledgements

We are grateful to Walter G Gonzalez and Antuca Callejas for comments on the manuscript.

## Additional information

### Funding

| Funder | Grant reference number | Author |
| --- | --- | --- |
| NIH Office of the Director | RO1MH116508 | Luis sanchez-guardado |

The funders had no role in study design, data collection and interpretation, or the decision to submit the work for publication.

### Author contributions

Luis Sánchez-Guardado, Conceptualization, Data curation, Formal analysis, Validation, Investigation, Visualization, Methodology, Writing—original draft, Writing—review and editing; Carlos Lois, Conceptualization, Resources, Data curation, Formal analysis, Supervision, Funding acquisition, Validation, Investigation, Visualization, Methodology, Writing—original draft, Project administration, Writing—review and editing

### Author ORCIDs

Luis Sánchez-Guardado (iD) https://orcid.org/0000-0001-5598-8608
Carlos Lois (iD) https://orcid.org/0000-0002-7305-2317

### Ethics

Animal experimentation: In all experiments, mice were handled according to the mice protocol (#1709) approved by the Institutional Animal Care and Use Committee (IACUC) of the California Institute of California.

### Decision letter and Author response

Decision letter https://doi.org/10.7554/eLife.46675.021
Author response https://doi.org/10.7554/eLife.46675.022

## Additional files

### Supplementary files

• Supplementary file 1. M/T cells clones in the accessory olfactory bulb (A) Summary of all M/T cells clones observed in the accessory olfactory bulb (AOB), indicating the number of cells found in the AOB and the fluorescent protein expressed by M/T cells in the AOB and main OB.
DOI: https://doi.org/10.7554/eLife.46675.018

• Transparent reporting form
DOI: https://doi.org/10.7554/eLife.46675.019

### Data availability

All data generated or analyzed during this study are included in the manuscript and supporting files.

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
