## [Decision Letter]

Thank you for submitting your article "Lineage does not regulate the connectivity of projection neurons in the mouse olfactory bulb" for consideration by *eLife*. Your article has been reviewed by three peer reviewers, one of whom is a member of our Board of Reviewing Editors, and the evaluation has been overseen by a Reviewing Editor and Catherine Dulac as the Senior Editor. The following individuals involved in review of your submission have agreed to reveal their identity: Stephen Liberles (Reviewer #1).

The reviewers have discussed the reviews with one another and the Reviewing Editor has drafted this decision to help you prepare a revised submission.

All reviewers noted that the work was interesting and the data generally convincing, but also indicated that additional experiments were needed to support principal conclusions. Reviewer comments are attached in their entirety below, but a few specific points are highlighted to help focus your revision efforts.

1) All reviewers asked for more validation that all mitral cells could be labeled by the approach. This is important given that dedicated subsets of mitral cells might be relevant for innate behaviors, and that such neurons might be born at different times, in different locations (dorsal/ventral), or not be marked in Nestin-Cre mice.

2) Reviewers 2/3 also ask for analysis of central projections, (reviewer 3 mentions the cortical amygdala as it has been implicated in innate behaviors). Are the central projections of different clonal populations indistinguishable from global mitral cell projections across higher order olfactory nuclei?

3) Reviewers also inquire about sister cells within the AOB, and whether sisters can be distributed among both the AOB and MOB. Is the logic underlying glomerulus connectivity of AOB sister cells distinct?

Please see all comments below.

*Reviewer #1:*

The authors use a genetic approach to visualize small groups of olfactory projection neurons with a putatively clonal origin. They find that neurons from the same putative clone can project to different glomeruli, and as such, they are likely relevant for different odor responses. These findings argue against lineage-based determinism underlying mitral cell identity. Given the large number of glomeruli, and the ability of the olfactory map to expand and contract with receptor number, lineage-based models do not seem likely from the get-go; that said, I am not sure whether lineage models have formally been challenged in the mouse. Certainly, the notion that glomerulus innervation occurs through an at least partially stochastic mechanism is an intuitive and attractive model, and the authors present data that is at least consistent with this idea. These findings are also consistent with recent work indicating that the receptor-guided spatial map in the olfactory bulb is discarded in the piriform cortex. It should be noted that it is possible that neuron specification occurs after time points of clonal expansion measured here, or in smaller subsets of neurons not detectable here (as the authors discuss).

I only have one technical point. The authors should provide additional evidence that clonal populations are being analyzed. They report ~20 projection neurons labeled per olfactory bulb, and 16/29 olfactory bulbs with labeled cells. Is only one clone (and thus one fluorophore) detected in all ~20 neurons from each of these 16 bulbs? In a table, provide the exact distribution of mitral cells with different fluorophores from each mouse as this information is essential for assessing the extent of clonality and whether the assumption at the end of the second paragraph of subsection “Labeling of progenitors of OB projection neurons” is valid. Conclusions depend on an absence of multiple independent labeling events.

*Reviewer #2:*

Summary:

Using a genetic strategy to sparsely label neural progenitors, Sánchez-Guardado and Lois found that sister olfactory bulb mitral/tufted cells innervate multiple glomeruli distributed across the olfactory bulb. This paper has relevance to the general problem of understanding how the bulb wires up, and implications for the organization of projections connecting the bulb to the cortex.

Experimental Concerns:

1) Previous work, such as Imamura et al., (2011, 2015), has shown that early-generated mitral cells (E10) cells are distributed more in dorsal domain of the olfactory bulb, whereas late-generated mitral (E12~13) cells are preferentially distributed in ventral domain. Given this, it is important to examine multiple timepoints for tamoxifen injection and ask if there are any timing-dependent changes in clonal distribution of mitral/tufted cells (e.g. test additional embryonic timepoints in addition to E10.5).

2) It is not clear if all the mitral/tufted cells are generated from Nestin-positive RGCs. This is important because otherwise the results shown here may miss subpopulations of more restricted or distributed sister mitral/tufted cells. Using Nestin-Cre or giving more tamoxifen can answer this.

3) The small number of clones (typically 1) labeled at a single time-point makes it hard to interpret their results as actually representative throughout the olfactory bulb. They authors should quantify or demonstrate to what extent their clones uniformly cover the olfactory bulb. In addition, more images of the clones themselves would be useful.

4) In addition to asking about the bulb, this work affords the opportunity to look at connectivity and organization in cortex. Given that prior work (e.g. Miyamichi et al., 2011; Sosulski et al., 2011) suggests that M/T cells innervating a single glomerulus project diffusely across olfactory regions, the authors could provide an explanation for these findings by examining the projection patterns of sister M/T cells, which they should be able to do by looking at downstream olfactory bulb targets of the clones labeled in this study.

5) The authors should examine if sister mitral/tufted cells were distributed across dorsal and ventral domains (as labeled with Nqo1 and OCAM, respectively) or they were specific to either domain. This could provide some information about how the distribution of sister cells is regulated.

6) The methods for their randomized NND analysis are unclear. The authors stated that "the distances were generated randomly with a normal distribution 230 between the longest and shortest distances observed between M/T cells." However, a normal distribution is specified by a mean and variance, neither of which are given. Furthermore, their randomization methods differ from those described previously (e.g. Gao et al., 2014), which aim to randomly sample within a given volume.

7) Is there any spatial bias in the position of labeled progenitor cells by this experiment? It would be better to show the quantification for this to make sure they labeled more or less throughout the entire pool of the progenitors in embryonic brain.

8) No discussion is made of AOB M/T cell clones. Do single clones contain sister cells in both the MOB and AOB?

9) Show all data points for Figure 3A and show some quantification of the distributions for all clones.

10) The citation in this sentence “This scenario could be expected for putative glomeruli responsive to relevant odors for survival, such as those responsive to predators or poisons, which require an innate and hardwired response of avoidance (Sosulski et al., 2011).” does not make sense.

Conceptual concerns:

1) Lack of discussion with respect to the timing OSN development: If it's known that OSNs innervate glomeruli after mitral/tufted cell differentiate and migrate to their positions in the olfactory bulb (e.g. Blanchart et al., 2006), would one necessarily expect that sister mitral cells innervate the same glomerulus?

2) If bulb neurons migrate tangentially, wouldn't one expect the clones to be distributed more than those in cortex? Likewise, as the authors state, since it's already known that the olfactory bulb has higher levels of direct neurogenesis rather than indirect neurogenesis via intermediate progenitors (Cárdenas et al., 2018), the smaller clone size of the olfactory bulb already suggests that these regions have different modes of development. Likewise, it remains unclear whether using a single time point for tamoxifen injection labels progenitors at equivalent stages of potency/development across these separate regions.

3) The authors did not show if sister mitral/tufted cells are connected to each other or connect to the same cells other than olfactory sensory neurons. The authors state that they "investigated whether lineage plays a role in the connectivity of mitral and tufted cells," however by connectivity here they only refer to connections with OSNs innervating a single glomerulus. Especially given the extent of intrabulbar connectivity, as well as recent reports of biases in connectivity among clonally-derived neocortical neurons, if the authors would like to claim that "lineage does not regulate the connectivity of projection neurons in the mouse olfactory bulb," which their Title reads, it is not sufficient to only show that sister mitral/tufted cells do not extend dendrites to the same glomerulus.

*Reviewer #3:*

In this report, the authors attempted to address an interesting issue in the mouse olfactory system, i.e., whether the lineage of mitral/tufted (M/T) cells determines the connectivity to their partner glomeruli within the olfactory bulb (OB). By using an elegant transgenic system, the authors sparsely labeled M/T cell progenitors to study whether sister M/T cells connect to the same glomerulus. They found that clonally-related M/T cells do not necessarily synapse with olfactory sensor neurons (OSNs) expressing the same odorant receptor. Based on this observation, the authors conclude that cell lineage does not determine the synaptic connectivity of M/T-cells with OSN axons.

In the paper, the experiments are well-designed and the results are mostly clear. As this paper presents basically negative results, adding some positive data as to a role of cell lineage in forming M/T-cell circuits would strengthen the paper. It will be quite interesting to examine whether the sister M/T cells send their axons to the same area in the olfactory cortex (OC). It will also strengthen the paper if the authors could discuss more about the connectivity of M/T cells regarding possible mechanisms that mediate partner matching with glomeruli. Specific comments are as follows:

1) In Figure 4, some examples are shown for sister M/T cells connecting to different glomeruli. Are there any differences in sister-cell distribution between the innate and non-innate OB regions?

2) Is it possible to determine when and where the sister cells are derived during embryonic development?

3) Even if sister cells do not connect to the same glomerulus, are there any shared characteristics and common features in their gene expression (particularly for axon guidance molecules), OC projection (particularly to the amygdala), and firing patterns?

4) It is worth examining M/T cells in the accessory OB where all M/T-cell circuits are hard-wired to mediate innate pheromone responses?

5) In the last part of Discussion section, the authors list interesting future questions. This paper would be significantly strengthened if any results could be added regarding these questions, particularly for connectivity to the amygdala.

---

## [Author Response]

All reviewers noted that the work was interesting and the data generally convincing, but also indicated that additional experiments were needed to support principal conclusions. Reviewer comments are attached in their entirety below, but a few specific points are highlighted to help focus your revision efforts.

We thank the editors and reviewers for their comments on the manuscript. We have performed the additional experiments they suggested and made other changes to the manuscript and figures to address their concerns. Finally, we have also made smaller changes to text to improve clarity and readability.

1) All reviewers asked for more validation that all mitral cells could be labeled by the approach. This is important given that dedicated subsets of mitral cells might be relevant for innate behaviors, and that such neurons might be born at different times, in different locations (dorsal/ventral), or not be marked in Nestin-Cre mice.

a) Are there different subsets of M/T cells born at different times with different characteristics?

We initially focused at embryonic day 10.5 because according to the existing literature, this is the time in which the majority of M/T cells are born (Hinds, 1968a, 1968b; Blanchart et al., 2006; Kim et al., 2011; Imamura et al., 2011). In our original submission, all data were derived from labeling E10.5 progenitors. The reviewers raised the question of whether clonally related cells born from later progenitors may produce sister M/T cells that innervate the same glomerulus. We analyzed a later time point of tamoxifen administration (E12.5) and have observed that sister M/T cells born from this E12.5 progenitors do not receive synaptic input from the same glomerulus. These new data are included in a new figure (Figure 4—figure supplement 1).

b) Are there progenitors that produce subsets of M/T cells preferentially destined to occupy the dorsal or ventral regions of the bulb?

We have analyzed the distribution of clones of M/T cells born both at E10.5 and E12.5. We have observed that at both of these time points there are clones preferentially located in the dorsal domain, in the ventral domain and clones with no preference for either dorsal or ventral domains. Moreover, when all clones are analyzed together, we did not observe any preference for distribution in the dorsal or ventral domains. These new data are now included in a new figure (Figure 3—figure supplement 1 (E10.5), and Figure 3—figure supplement 4 (E12.5))

c) Are all M/T cells labeled in Nestin-cre mice?

We agree that these are important questions and we have performed additional experiments to address them. Below we will reply to these questions in order:

To address this question, we have performed two complementary experiments.

First, according to the literature, progenitors for M/T cells are located in the rostralmost region of the forebrain, also called the presumptive olfactory bulb. As suggested by one of the reviewers we crossed the *Nestin-cre* mouse with the *Ai9* mouse strain, a highly sensitive loxP-cre reporter that expresses tdTomato upon cre mediated recombination. When we analyzed these animals at E10.5 we observed that all neuronal progenitors (identified by the expression of Pax6 and RC2 markers) are also positive for tdTomato. This observation indicates that the *Nestin-cre* can activate recombination of a loxP-reporter in all neuronal progenitors located in the presumptive OB at the time when we labeled the clones of M/T cells that we analyzed in the OB. These new data are presented in a new figure (Figure 1—figure supplement 2)

Second, we analyzed the expression of the tdTomato marker in the OB in a postnatal day 7 mice carrying the *Nestin-cre* and *Ai9* reporter alleles. We observed that all M/T cells analyzed (identified by the expression of the M/T cells marker Tbx21) are also positive for the tdTomato marker. These new data are presented in a new figure (Figure 1—figure supplement 3)

2) Reviewers 2/3 also ask for analysis of central projections, (reviewer 3 mentions the cortical amygdala as it has been implicated in innate behaviors). Are the central projections of different clonal populations indistinguishable from global mitral cell projections across higher order olfactory nuclei?

Indeed, this is an interesting question, and we included a brief discussion of this possibility in our original submission. The reason why we only mentioned this as a possibility, but we decided not to present any data in this regard is because with the currently available transgenic mice it is not possible to reliably address this issue. Recently, several works have traced the final destination of axons originating from M/T cells locally labeled in the OB (Sosulski et al., 2011, Ghosh et al., 2011, Igarashi et al., 2012). Although there are transgenic mice that can be used to selectively label neocortical progenitors (such as *Emx1-CreER^T2^*), currently there are no transgenic mice capable of selective labeling of M/T progenitors. Because of this limitation, to label progenitors of M/T cells we used the *Nestin-CreER^T2^* mice that labels all neuronal progenitors throughout the brain. Therefore, when we induce cre recombination in these mice we label, not only M/T progenitors but also many other progenitors in other brain regions, including the olfactory cortex and other brain regions with neurons whose axons project into the olfactory cortex. When we attempted to analyze the trajectories of axons in the olfactory cortex we observed some local neurons labeled in the olfactory cortex (see Author response image 1, illustrating a labeled neuron in the olfactory cortex). These neurons have axons that extended locally within the olfactory cortex, and this makes it extremely challenging to discern whether a given labeled axon in the olfactory cortex belongs to a M/T cell from the OB, or to local neurons in the olfactory cortex. Thus, we did not attempt to trace the destination of axons in the olfactory cortex because we could not unambiguously identify the cells from which they originated. As an example, we attach confocal images from two brains as examples of axons in the olfactory cortex that illustrate this challenge.

**Author response image 1. respfig1:** Axons in olfactory cortical regions. (A-F) Confocal images of the axons in the lateral olfactory tract (LOT), in two different brains, at different anterior-posterior levels of the brain in coronal sections. (A-C) Axons in the LOT at the anterior olfactory nucleus and (D-F) piriform cortex in coronal sections from a brain with M/T cells labeled in the OB. Scale bar in F is 100 µm and applies to A-F. Orientation of brain: D, dorsal; M, medial.

3) Reviewers also inquire about sister cells within the AOB, and whether sisters can be distributed among both the AOB and MOB. Is the logic underlying glomerulus connectivity of AOB sister cells distinct?

This is a very interesting question, especially because the AOB is thought to process smells involved in innate behaviors, which could be genetically programmed. In our initial submission we decided to exclusively focus on the relationship between lineage and connectivity for M/T cells in the MOB, because identifying the connectivity of dendrites in the MOB is unambiguous, but the anatomical characteristics of the AOB make it a less reliable system where to study this question. In the MOB, M/T cells have a single apical dendrite that projects into a single, well-defined glomerulus. The AOB has a critical caveat to study questions related to synaptic connectivity because its glomeruli are not well defined, and thus, are difficult to identify. In addition, we also observed that whereas the number of cells per clone in the MOB was (on average) 20 cells, for the AOB the clone size was much smaller (between 1-3 per AOB). Taking into account these limitations we now include these data in the revised manuscript. However, we think that given these caveats we thought that it would be important to temper any interpretation, as follows: “Although the small number of labeled AOB M/T cells does not allow us to draw any firm conclusions, we did not find any M/T cells whose apical dendrites innervated the same glomerulus (Figure 4—figure supplement 2), similar to what we observed in the MOB”.

Please see all comments below.Reviewer #1:[…] I only have one technical point. The authors should provide additional evidence that clonal populations are being analyzed. They report ~20 projection neurons labeled per olfactory bulb, and 16/29 olfactory bulbs with labeled cells. Is only one clone (and thus one fluorophore) detected in all ~20 neurons from each of these 16 bulbs? In a table, provide the exact distribution of mitral cells with different fluorophores from each mouse as this information is essential for assessing the extent of clonality and whether the assumption at the end of the second paragraph of subsection “Labeling of progenitors of OB projection neurons” is valid. Conclusions depend on an absence of multiple independent labeling events.

We agree that this is a valuable addition to the ms., and we have now included all these data in two new supplementary figures (Figure 3—figure supplement 1 and Figure 3—figure supplement 4). In these figures we show all the clones obtained by labeling at E10.5 and E12.5, and we indicate the following: (i) the 3D reconstructions of the OBs illustrating the distribution of each labeled M/T cell through the bulb, including the potential bias towards a dorsal or ventral domains, (ii) the number of cells labeled in each putative clone, and (iii) the fluorophore of each clone, including those bulbs that contain cells labeled with 2 fluorophores.

Reviewer #2:[…] Experimental Concerns:1) Previous work, such as Imamura et al., (2011, 2015), has shown that early-generated mitral cells (E10) cells are distributed more in dorsal domain of the olfactory bulb, whereas late-generated mitral (E12~13) cells are preferentially distributed in ventral domain. Given this, it is important to examine multiple timepoints for tamoxifen injection and ask if there are any timing-dependent changes in clonal distribution of mitral/tufted cells (e.g. test additional embryonic timepoints in addition to E10.5).

As requested by this reviewed we have performed new experiments in which TMX was administered at E12.5. These new data are now shown in a new figure (Figure 3—figure supplement 4). Again, as suggested by the reviewer we have analyzed the distribution of the labeled M/T cells to investigate for a possible bias towards the dorsal or ventral regions of the OB from the clones labeled at E10.5 and E12.5. These new data are shown in two figures: Figure 3—figure supplement 1 and Figure 3—figure supplement 4.

We have observed that at both of these time points there are clones preferentially located in the dorsal regions, in the ventral regions and clones with no preference for either dorsal or ventral regions. Moreover, when all E10.5 clones are analyzed together, we do not observe any clear preference for distribution in the dorsal or ventral regions. For the clones labeled at E12.5 there is tendency towards labeled M/T cells to occupy more ventral regions of the OB (consistent with Imamura et al. data), but the statistical analysis reveals that these differences do not reach significance.

Imamura et al., (2011) show that E10 generated M/T cells have a bias towards the rostral regions of the OB, in an approximate ratio of 1 (dorsal) to 0.75 ventral. Similarly, the M/T cells generated at E12 have an approximate bias of 0.6 (dorsal) to 1 (ventral). In their experiments, they labeled M/T progenitors with thymidine analogs (brdu, IDU and CiDU), and presumably their samples included thousands of cells which allowed them to perform a robust statistical analysis. For our experimental design it was crucial to label a small number of clones per bulb (ideally a single clone per bulb), which resulted in a small number of total cells labeled, even combining all the animals studied. As mentioned above, our analysis of progenitors labeled at E12.5 may suggest that those late-generated clones could have a bias ventral as demonstrated by Imamura et al., (2011). We would like to point out that the main objective of clonal analysis is to identify lineage relationships between cells, or their distribution within a single clone. However, the small sample numbers provided by clonal analysis are not well suited to provide information about the distribution of all the neurons in a cohort born in a given day, a type of data for which the birthday dating method used by Imamura et al., (2011) is best suited.

2) It is not clear if all the mitral/tufted cells are generated from Nestin-positive RGCs. This is important because otherwise the results shown here may miss subpopulations of more restricted or distributed sister mitral/tufted cells. Using Nestin-Cre or giving more tamoxifen can answer this.

Of the 2 strategies suggested by this reviewer, we chose to use the *Nestin-cre* x *Ai9* because increasing the dose of TMX into pregnant females triggers abortions (see Danielian et al., 1998; Ved et al., 2019). As suggested by this reviewer, we crossed the driver *Nestin-Cre* mouse with the reporter *Ai9* mouse. In postnatal animals (P7), all M/T cells in the *Nestin-cre::Ai9* mice were labeled in the MOB (Figure 1—figure supplement 3).

3) The small number of clones (typically 1) labeled at a single time-point makes it hard to interpret their results as actually representative throughout the olfactory bulb.

1a) Labeling of single clones per bulb:

Labeling a single clone per bulb is necessary to increase the likelihood that all the sister cells labeled belong to a single clone. In our experiments we used the Confetti mice because this strain can produce clones in 4 different distinguishable markers. With this strain, even if 2 clones are labeled in an olfactory bulb and each clone is labeled with a different color, one can infer that these are 2 independent clones, originating from individual progenitors. However, if several clones are labeled per olfactory bulb (for example, 5 clones in one bulb) some of the clones will be labeled with the same color thus making it impossible to discern how many individual clones are labeled.

1b) Labeling at different time points:

As suggested by this reviewer we have performed additional experiments to label clones at a later time point, E12.5. We have observed that, as expected, the number of cells per clone is smaller at E12.5 compared with E10.5. However, the characteristics of the clones are similar in these 2 time points. As we mentioned in response to point 1 from the reviewer, both at E10.5 and E12.5 we find clones where the sister cells are mostly located ventrally or dorsally, but we also find clones where the sister cells are distributed throughout all the bulb regions without any clear bias. In aggregate, when all clones are analyzed together, we do not observe any clear preference for distribution in the dorsal or ventral regions. These new data are shown in two new figures (Figure 3—figure supplement 1 and Figure 3—figure supplement 4).

1c) Are the clones observed representative?:

We mention in the discussion that it is theoretically possible that there may exist some clones where the sister cells project to the same glomerulus, that those clones may be rare and that we could we have not been able to label those putative clones in our experiments. However, the analysis of the clones that we have studied reveals several commonalities: (i) the M/T cells are not concentrated in defined regions of the bulb, and instead, they are distributed throughout large volumes, (ii) none of the sister M/T cells in one clone project to the same glomerulus, (iii) the size of the clones ranges between 12 and 33 M/T cells, and between 3 and 22 M/T cells, when labeling at E10.5 and E12.5, respectively.

They authors should quantify or demonstrate to what extent their clones uniformly cover the olfactory bulb. In addition, more images of the clones themselves would be useful.

As indicated above we have performed this analysis, and included these data in 2 new figures where we incorporated the 3D reconstruction of the 14 clones detected in our E10.5 experiment (Figure 3—figure supplement 1) and the 9 clones in the E12.5 experiment (Figure 3—figure supplement 4).

4) In addition to asking about the bulb, this work affords the opportunity to look at connectivity and organization in cortex. Given that prior work (e.g. Miyamichi et al., 2011; Sosulski et al., 2011) suggests that M/T cells innervating a single glomerulus project diffusely across olfactory regions, the authors could provide an explanation for these findings by examining the projection patterns of sister M/T cells, which they should be able to do by looking at downstream olfactory bulb targets of the clones labeled in this study.

As indicated in the general comment to all reviewers, this is an interesting question, and we included a brief discussion of this possibility in our original submission. The reason why we only mentioned this as a possibility but we decided not to present any data in this regard is because with the currently available transgenic mice it is not possible to reliably address this issue. Recently, several works have traced the final destination of axons originating from M/T cells locally labeled in the OB (Sosulski et al., 2011, Ghosh et al., 2011, Igarashi et al., 2012). Although there are transgenic mice that can be used to selectively label neocortical progenitors (such as *Emx1-CreER^T2^*), currently there are no transgenic mice capable of selective labeling of M/T progenitors. Because of this limitation, to label progenitors of M/T cells we used the *Nestin-CreER^T2^* mice that labels all neuronal progenitors throughout the brain. Therefore, when we induce cre recombination in these mice we label, not only M/T progenitors but also many other progenitors in other brain regions, including the olfactory cortex and other brain regions with neurons whose axons project into the olfactory cortex. When we attempted to analyze the trajectories of axons in the olfactory cortex we observed some local neurons labeled in the olfactory cortex (see reviewer Figure 1F, illustrating a labeled neuron in the olfactory cortex). These neurons have axons that extended locally within the olfactory cortex, and this makes it extremely challenging to discern whether a given labeled axon in the olfactory cortex belongs to a M/T from the OB, or to local neurons in the olfactory cortex. Thus, we did not attempt to trace the destination of axons in the olfactory cortex because we could not unambiguously identify the cells from which they originated. As an example, we attach confocal images from two brains as examples of axons in the olfactory cortex that illustrate this challenge.

5) The authors should examine if sister mitral/tufted cells were distributed across dorsal and ventral domains (as labeled with Nqo1 and OCAM, respectively) or they were specific to either domain. This could provide some information about how the distribution of sister cells is regulated.

We analyzed the distribution of M/T cells in the OB based on the dorsal and ventral domains as revealed by the NQO1 and OCAM markers (Gussing and Bohm, 2004; Yoshihara et al., 1997). These data are now presented in a new figure (Figure 3—figure supplement 1L and Figure 3—figure supplement 4J). We did not observe any preference for any given OB domain when we analyzed all the clones distribution together (Figure 3—figure supplement 1L and Figure 3—figure supplement 4J).

6) The methods for their randomized NND analysis are unclear. The authors stated that "the distances were generated randomly with a normal distribution 230 between the longest and shortest distances observed between M/T cells." However, a normal distribution is specified by a mean and variance, neither of which are given. Furthermore, their randomization methods differ from those described previously (e.g. Gao et al., 2014), which aim to randomly sample within a given volume.

We apologize for this mistake, because in our original submission we randomize the data based on an assumed normal distribution of the cell location. As the reviewer indicates, this is not appropriate, and we have now corrected this mistake and we recalculated the data based on their location within a volume (in 3D). In the update Figure 3E, the random data were generated based on the volume of the external plexiform layer of one of the OB analyzed, and the volume of a neocortex clone with a volume closer to the average neocortical clone volume. To illustrate how we calculated the randomization of the data, we have included a 3D diagram in Author response image 2. Using these new criteria, we observed that the results are identical with those presented in the original submission, which were calculated with different criteria): (i) there are no differences when comparing the characteristics of the different “experimental” clones found in the olfactory bulb, and (ii) there are significant differences when the “experimental” clones are compared with the “simulated” clones generated by randomizing the “experimental” dataset, both in the OB and neocortex

**Author response image 2. respfig2:** Volume for random dataset in the olfactory bulb and neocortex. (A-C) 3D reconstruction from clones observed in OB (A-B) and neocortex (**D**) on our E10.5 experiment. (D-F) 3D reconstruction used to generate the random data set in the OB (D-E) and the neocortex (**F**).

7) Is there any spatial bias in the position of labeled progenitor cells by this experiment? It would be better to show the quantification for this to make sure they labeled more or less throughout the entire pool of the progenitors in embryonic brain.

The information that can be obtained from clonal analysis regards the relationship between the sister cells that are derived from a single progenitor. The main objective of our study was to investigate whether “sister” M/T cells receive synaptic input from a single glomerulus, and our data indicates that they do not. However, as in any clonal analysis experiment, we cannot say anything about the position of the specific progenitor that generated those mature cells, because those progenitors disappear when they differentiate.

Based on the current literature, it is generally assumed that the M/T cell progenitors for the olfactory bulb are located in the rostral part of the forebrain, in the so-called presumptive olfactory bulb. Crossing the driver mouse *Nestin-Cre* with the reporter mouse *Ai9*, we observed at E10.5 that progenitors tdTomato+ co-expressed the progenitor markers RC2 and Pax6 in the presumptive OB (Figure 1—figure supplement 2). Based on these results, any putative progenitor cells in the OB primordium has the potential to be labeled by the TMX administration time. However, as mentioned above, when we analyze the characteristics of the mature M/T cells in a clone we cannot infer anything regarding the position of the progenitor from which they originated.

8) No discussion is made of AOB M/T cell clones. Do single clones contain sister cells in both the MOB and AOB?

The main objective of our study is to investigate the relationship between lineage and connectivity, and accordingly we decided to focus our analysis on the connectivity of M/T cells in the main olfactory bulb. However, we agree that lineage relationships between the main and accessory olfactory bulb is an interesting question, and we now include this new paragraph in the main text of the manuscript to discuss this issue:

“Our experiments were designed to investigate the relationship between lineage and connectivity in the main olfactory bulb (MOB). […] Further experiments will be required to clarify these questions.”

9) Show all data points for Figure 3A and show some quantification of the distributions for all clones.

As suggested by this reviewer, we now include all data points in this plot (Figure 3A). We did the same for the new E12.5 data (Figure 3—figure supplement 4B)

10) The citation in this sentence “This scenario could be expected for putative glomeruli responsive to relevant odors for survival, such as those responsive to predators or poisons, which require an innate and hardwired response of avoidance (Sosulski et al., 2011).” does not make sense.

In the Discussion section we wrote: “This scenario could be expected for putative glomeruli responsive to relevant odors for survival, such as those responsive to predators or poisons, which require an innate and hardwired response of avoidance (Sosulski et al., 2011).” The reason why we decided to include this reference is because the take-home message from the Sosulski article is that the connectivity between the olfactory bulb and the olfactory cortex varies from animal to animal, presumably to process smells whose interpretation has to be learned. In contrast, the connectivity between the bulb and the amygdala is stereotyped, and this is thought to mediate smells associated with innate behaviors (such as predator or poison avoidance). For example, in the Sokulski et al., (2011) Abstract, they mention “The identification of a distributive pattern of projections to the piriform and stereotyped projections to the amygdala provides an anatomical context for the generation of learned and innate behaviours.” We thought that it was appropriate to give credit to a work that studied these questions, and emphasized how differences in connectivity could explain the differences between learned and innate behaviors.

Conceptual concerns:1) Lack of discussion with respect to the timing OSN development: If it's known that OSNs innervate glomeruli after mitral/tufted cell differentiate and migrate to their positions in the olfactory bulb (e.g. Blanchart et al., 2006), would one necessarily expect that sister mitral cells innervate the same glomerulus?

We agree that these are important issues and we have now extended the Discussion section to address these questions (see subsection “Synaptic input of sister M/T cells”).

2) If bulb neurons migrate tangentially, wouldn't one expect the clones to be distributed more than those in cortex?

The fact that immature M/T cells migrate tangentially does not necessarily mean that sister M/T cells would be dispersed. For example, one could imagine a scenario in which a group of sister M/T cells migrate long distances tangentially to reach a specific domain of the bulb, but that all those sister would end up close to each other.

Likewise, as the authors state, since it's already known that the olfactory bulb has higher levels of direct neurogenesis rather than indirect neurogenesis via intermediate progenitors (Cárdenas et al., 2018), the smaller clone size of the olfactory bulb already suggests that these regions have different modes of development. Likewise, it remains unclear whether using a single time point for tamoxifen injection labels progenitors at equivalent stages of potency/development across these separate regions.

As requested by this reviewer we performed a new experiment where we administered TMX to a 12 days pregnant female (E12.5). We have found that on average the number of cells per clone at E12.5 (around 9 cells) is smaller than at E10.5 (around 22 cells). These data suggest that in our experiments we are labeling the same M/T progenitors at E10.5 and E12.5, but that at a later time points those progenitors have already produced much of its progeny. (see Figure 3—figure supplement 4).

3) The authors did not show if sister mitral/tufted cells are connected to each other or connect to the same cells other than olfactory sensory neurons.

We would like to point out that our main interest was to investigate the possible relationship between lineage and sensory synaptic input, which we believe is a central question in developmental neuroscience. The relationship between lineage and synaptic connectivity has been mostly studied in just two systems: (a) the *Drosophila* antennal lobe where it has been shown that lineage determines the synaptic input of the principal neurons (the equivalent of M/T cells), and (b) the pyramidal neurons in the neocortex, where it has been suggested that sister neurons are preferentially connected to each other. To further investigate the possible relationship between lineage and synaptic input, we focused on the connectivity of M/T cells because the anatomical organization of the olfactory bulb, with its well-defined glomeruli is ideally suited to study this question.

The authors state that they "investigated whether lineage plays a role in the connectivity of mitral and tufted cells," however by connectivity here they only refer to connections with OSNs innervating a single glomerulus. Especially given the extent of intrabulbar connectivity, as well as recent reports of biases in connectivity among clonally-derived neocortical neurons, if the authors would like to claim that "lineage does not regulate the connectivity of projection neurons in the mouse olfactory bulb," which their Title reads, it is not sufficient to only show that sister mitral/tufted cells do not extend dendrites to the same glomerulus.

We agree with this comment. With our experiments we can only conclude that sister M/T cells do not receive sensory input from the same glomeruli. Accordingly, we have changed the Title of the manuscript which now reads as follows: “Lineage does not regulate the sensory synaptic input of projection neurons in the mouse olfactory bulb”.

Reviewer #3:[…] In the paper, the experiments are well-designed and the results are mostly clear. As this paper presents basically negative results, adding some positive data as to a role of cell lineage in forming M/T-cell circuits would strengthen the paper. It will be quite interesting to examine whether the sister M/T cells send their axons to the same area in the olfactory cortex (OC).

As indicated in the general comment to all reviewers, this is an interesting question, and we included a brief discussion of this possibility in our original submission. The reason why we only mentioned this as a possibility, but we decided not to present any data in this regard is because with the currently available transgenic mice it is not possible to reliably address this issue. Recently, several works have traced the final destination of axons originating from M/T cells locally labeled in the OB (Sosulski et al., 2011, Ghosh et al., 2011, Igarashi et al., 2012). Although there are transgenic mice that can be used to selectively label neocortical progenitors (such as *Emx1-CreER^T2^*), currently there are no transgenic mice capable of selective labeling of M/T progenitors. Because of this limitation, to label progenitors of M/T cells we used the *Nestin-CreER^T2^* mice that labels all neuronal progenitors throughout the brain. Therefore, when we induce cre recombination in these mice we label, not only M/T progenitors but also many other progenitors in other brain regions, including the olfactory cortex and other brain regions with neurons whose axons project into the olfactory cortex. When we attempted to analyze the trajectories of axons in the olfactory cortex we observed some local neurons labeled in the olfactory cortex (see reviewer Figure 1F, illustrating a labeled neuron in the olfactory cortex). These neurons have axons that extended locally within the olfactory cortex, and this makes it extremely challenging to discern whether a given labeled axon in the olfactory cortex belongs to a M/T from the OB, or to local neurons in the olfactory cortex. Thus, we did not attempt to trace the destination of axons in the olfactory cortex because we could not unambiguously identify the cells from which they originated. As an example, we attach confocal images from two brains as examples of axons in the olfactory cortex that illustrate this challenge.

It will also strengthen the paper if the authors could discuss more about the connectivity of M/T cells regarding possible mechanisms that mediate partner matching with glomeruli.

We agree that this is an important issue, and have added a long paragraph discussing this issue immediately after presenting our data (see subsection “Synaptic input of sister M/T cells”).

Specific comments are as follows:1) In Figure 4, some examples are shown for sister M/T cells connecting to different glomeruli. Are there any differences in sister-cell distribution between the innate and non-innate OB regions?

This is an interesting question. Some works have suggested that the dorsal and ventral regions of the olfactory bulb preferentially process innate and learned odorants, respectively. We have analyzed the distribution of M/T cells clones born both at E10.5 and E12.5. We have observed that at both of these time points there are clones preferentially located in the dorsal domain, in the ventral domain, and clones with no preference for either dorsal or ventral domains. Moreover, when all clones are analyzed together, we do not observe any preference for M/T cells distribution in the dorsal or ventral domains. These new data are now included in a new figure (Figure 3—figure supplement 1 (E10.5), and Figure 3—figure supplement 4 (E12.5))

2) Is it possible to determine when and where the sister cells are derived during embryonic development?

The information that can be obtained from clonal analysis regards the relationship between the sister cells that are derived from a single progenitor. The main objective of our study was to investigate whether sister M/T cells receive synaptic input from a single glomerulus, and our data indicates that they do not. However, as in any clonal analysis experiment, we cannot say anything about the position of the specific progenitor that generated those mature cells, because those progenitors disappear as such when they differentiate.

3) Even if sister cells do not connect to the same glomerulus, are there any shared characteristics and common features in their gene expression (particularly for axon guidance molecules), OC projection (particularly to the amygdala), and firing patterns?

We would like to point out that our main interest was to investigate the possible relationship between lineage and synaptic input, which we believe is a central question in developmental neuroscience. This question has been mostly studied in just two systems: (a) the *Drosophila* antennal lobe where it has been shown that lineage determines the synaptic input of the principal neurons (the equivalent of M/T cells), and (b) the pyramidal neurons in the neocortex, where it has been suggested that sister neurons are preferentially connected to each other. To further investigate the possible relationship between lineage and synaptic input, we focused on the connectivity of M/T cells because the anatomical organization of the olfactory bulb, with its well-defined glomeruli is ideally suited to study this question.

We agree that investigating whether sister M/T cells share other properties (gene expression, axonal projections or firing patterns) are interesting questions, and we mention some of these scenarios in our Discussion section. However, we believe that those questions, although interesting, are not the main question of this work.

4) It is worth examining M/T cells in the accessory OB where all M/T-cell circuits are hard-wired to mediate innate pheromone responses?

This is a very interesting question, especially because the AOB is thought to process smells involved in innate behaviors, which could be genetically programmed. In our initial submission we decided to exclusively focus on the relationship between lineage and connectivity for M/T cells in the MOB, because identifying the connectivity of dendrites in the MOB is unambiguous, but the anatomical characteristics of the AOB make it a less reliable system where to study this question. In the MOB, M/T cells have a single dendrite that projects into a single, well-defined glomerulus. The AOB has a critical caveat to study questions related to synaptic connectivity because its glomeruli are not well defined, and thus, are difficult to identify. In addition, we also observed that whereas the number of cells per clone in the MOB was (on average) 20 cells, for the AOB we clone size was much smaller (between 1-3 per AOB). Taking into account these limitations we now mention these data in the revised manuscript as follows: “Although the small number of labeled AOB M/T cells does not allow us to draw any firm conclusions, we did not find any M/T cells whose apical dendrites innervated the same glomerulus (Figure 4—figure supplement 2), similar to what we observed in the MOB”.

5) In the last part of Discussion section, the authors list interesting future questions. This paper would be significantly strengthened if any results could be added regarding these questions, particularly for connectivity to the amygdala.

We agree, and we attempted to perform this experiment. However, as we explained above the currently available transgenic do not allow us to reliably trace the projection of the axons of sister M/T into the olfactory cortex and amygdala.